# Regional and Seasonal Distributions of *N*-Nitrosodimethylamine (NDMA) Concentrations in Chlorinated Drinking Water Distribution Systems in Korea

**Sunyoung Park [1]** ⬤, **Sungjin Jung [1]** and **Hekap Kim [2],***

[1] Department of Environmental Science, Kangwon National University, Chuncheon, Gangwon-do 24341, Korea; tj54795@naver.com (S.P.); jsjbin878@naver.com (S.J.)

[2] School of Natural Resources and Environmental Science, Kangwon National University, Chuncheon, Gangwon-do 24341, Korea

* Correspondence: kimh@kangwon.ac.kr; Tel.: +82-33-250-8577

**Abstract:** Volatile *N*-Nitrosamines (NAs), including *N*-nitrosodimethylamine (NDMA), an emerging contaminant in drinking water, have been reported to induce cancer in animal studies. This study aims to investigate the regional and seasonal distributions of the concentrations of NDMA, one of the most commonly found NAs with high carcinogenicity, in municipal tap water in Korea. NDMA in water samples was quantitatively determined using high-performance liquid chromatography-fluorescence detection (HPLC-FLD) as a 5-dimethylamino-1-naphthalenesulfonyl (dansyl) derivative after optimization to dry the SPE adsorbent and remove dimethylamine prior to derivatization. Tap water samples were collected from 41 sites in Korea, each of which was visited once in summer and once in winter. The average (±standard deviation) NDMA concentration among all the sites was 46.6 (±22.7) ng/L, ranging from <0.13 to 80.7 ng/L. Significant NDMA differences in the regions, excluding the Jeju region, were not found, whereas the average NDMA concentration was statistically higher in winter than in summer. A multiple regression analysis for the entire data set indicated a negative relationship between NDMA concentration and water temperature. High levels of NDMA in Korea may pose excessive cancer risks from the consumption of such drinking water.

**Keywords:** cancer risk; distribution; drinking water; *N*-nitrosodimethylamine; regional; seasonal

## 1. Introduction

*N*-Nitrosamines (NAs) in municipal drinking water have been reported as a group of disinfection byproducts (DBPs) formed by chloramination (chlorination of water containing ammonia or nitrite) and ozonation [1]. Among them, six NAs, including *N*-nitrosodimethylamine (NDMA), *N*-nitrosomethylethylamine (NMEA), *N*-nitrosodiethylamine (NDEA), *N*-nitrosodipropylamine (NDPA), *N*-nitrosodibutylamine (NDBA), and *N*-nitrosopyrrolidine (NPYR) have been classified as probable human carcinogens (group B2) and designated as contaminants under the second Unregulated Contaminant Monitoring Rule (UCMR 2) by the United States Environmental Protection Agency (EPA) [2].

In particular, NDMA has been of the greatest concern because it is the most frequently found NA in drinking water and poses an extremely high carcinogenic risk from oral exposure with a drinking water unit risk (DWUR) of $1.4 \times 10^{-3}$ per µg/L [3]. This value is ~1000 times greater than those of bromodichloromethane ($1.8 \times 10^{-6}$ per µg/L) [4] and dichloroacetic acid ($1.4 \times 10^{-6}$ per µg/L) [5], which are frequently found at µg/L levels in chlorinated drinking water. However, NDEA has been found

much less frequently and in lower concentrations than NDMA [6,7], even though its carcinogenic potential is reported to be the greatest among the NAs (DWUR of $4.3 \times 10^{-3}$ per µg/L) [8]. Furthermore, diethylamine (DEA), which is a major organic precursor to NDEA, is rarely found in drinking water or surface water samples in the study sites [9]. Therefore, this study is limited to NDMA.

NDMA concentrations vary significantly according to region and season. A nationwide survey of source and finished water samples collected in Japan indicated that NDMA concentrations were lower in finished water samples than those in source water samples and that those in the finished water samples were higher in winter than in summer [10]. However, NDMA was detected to be above the method detection limit (MDL) in only a few finished samples (25.4%, 15/59) and had a maximum concentration of 10 ng/L [10]. To the contrary, a study conducted in Spain [11] indicated that NDMA concentration increased throughout the distribution system following chlorination. This study also indicated higher NDMA concentrations in winter than in summer and fall (3.2–20 ng/L vs. 1.5–2.5 ng/L and 0.89–9.2 ng/L in the distribution system, respectively) [11]. A nationwide survey conducted in Korea from 2013 to 2015 indicated a very low detection rate (8.1%) and a maximum concentration of 13.0 ng/L [12]. Studies conducted in China found similar values in finished water and tap water samples with a maximum concentration of 13.9 ng/L [13,14]. However, some surveys carried out in Canada and the United States revealed much higher concentrations of NDMA in finished water and tap water (up to 630 ng/L) than the other aforementioned studies [15–17].

The NDMA concentrations measured in the plants may differ significantly from those of the water in the faucets from which people consume water supplied through water distribution systems. Higher concentrations of NDMA typically corresponded with increased residence time in distribution systems, and higher NDMA concentrations were observed in the faucets than in the plants [11,18,19]. Therefore, from a public health perspective, it would be beneficial to measure the concentrations of NDMA in drinking water collected from the faucets from which water is consumed by the public with/without additional processing (e.g., heating/boiling and further treatment through water purifiers). Nonetheless, human health risk assessments for most DBPs including NDMA are commonly conducted using the data obtained for water at treatment plants. This may lead to the underestimation of possible health effects caused by the exposure via oral consumption, inhalation, and dermal absorption. This is also true of Korea, where it appears that NDMA is not a great concern because of low excess cancer risks ($<10^{-6}$) estimated using the treatment plant data.

NDMA concentrations vary according to the season and are typically higher in winter than in other seasons [10,11,18]. A higher concentration of NDMA in winter appears to be attributed to less effective pre-chlorination, which is used to destroy NDMA precursors, including dimethylamine (DMA), at lower temperatures [19].

To quantitatively determine the concentration of NDMA in water, pretreatment procedures, including NDMA adsorption onto solid particles, sorbent drying, and solvent elution steps, are crucial. Aqueous NDMA is commonly adsorbed onto coconut charcoal [20,21], Ambersorb 572 [6,17], and Carboxen 572 [22]. The compound is then eluted with an organic solvent, such as dichloromethane (DCM) or a mixture of DCM and methanol, following adsorbent drying under vacuum or purging with inert gas (e.g., $N_2$). The eluate is concentrated and then analyzed for NDMA using either gas chromatography (GC) or high-performance liquid chromatography (HPLC) following derivatization.

In the above procedures, the chosen adsorbent and drying method are critical. NDMA might form on the surface of activated carbon from secondary amines in the presence of nitrogen and oxygen during sample loading and adsorbent drying [23,24]. This suggests that the rapid drying of the adsorbent with or without minimal contact with atmospheric gases is necessary to avoid the artifact formation of NDMA during the pretreatment process. Adsorbent drying is typically performed with a vacuum pump [20,22,25,26] or nitrogen gas flow [14,21]. However, detailed descriptions of vacuum pressure, flow rate, and/or drying time have not been made. Thus, excessive or insufficient drying may lead to the loss or incomplete extraction of NDMA, respectively.

When HPLC coupled with fluorescence detection (FLD) is used to determine NAs, including NDMA, they are most frequently converted to fluorescent 5-dimethylamino-1-naphthalenesulfonyl (dansyl) derivatives [22,27–29]. However, this method requires the removal of residual secondary amines, including DMA, which are organic precursors to NAs from the eluate prior to derivatization, because drinking water typically contains a few μg/L levels of secondary amines [9,30,31]. Otherwise, NA concentrations can be overestimated because of the additional formation of dansyl derivatives as a result of the reactions of secondary amines contained in the water itself, which are not formed from NAs by denitrosation with dansyl chloride.

In this study, an analytical method for determining NDMA in water using HPLC-FLD is established following the optimization of sorbent-drying conditions and the removal of secondary amines. Thereafter, the regional and seasonal distributions of the NDMA concentrations in tap water samples collected nationwide from 41 sites in Korea in summer and winter are investigated.

## 2. Materials and Methods

### 2.1. Chemicals and Reagents

NDMA, *N*-nitrosomethylbutylamine (NMBA, a surrogate), DMA, sodium thiosulfate ($Na_2S_2O_3$), and Carboxen® 572 were purchased from Sigma-Aldrich (St. Louis, MO, USA). Acetone, acetonitrile ($CH_3CN$), DCM, and methanol were obtained from Honeywell Burdick & Jackson (Muskegon, MI, USA), and NaOH, $NaHCO_3$, $Na_2SO_4$, and glacial acetic acid were purchased from Daejeong Chemicals & Metals (Siheung, Gyeonggi-do, Korea). Dansyl chloride and 48% hydrobromic acid were purchased from Calbiochem (San Diego, CA, USA) and Wako Pure Chemical Industries, Ltd. (Osaka, Gansai, Japan), respectively. Silica gel blue was purchased from Showa (Saitama, Japan).

A reagent for the denitrosation of NDMA to dimethylamine was prepared by diluting 1 mL of a 48% HBr solution to 10 mL with glacial acetic acid. A dansylating reagent was made by dissolving 25 mg of dansyl chloride in acetone and diluting it to 50 mL. A pH 10.5 buffer solution was prepared by dissolving 0.6 g of NaOH and 2.0 g $NaHCO_3$ in water and diluting it to 50 mL. All reagents were stored in a refrigerator at 4 °C and used within 2 weeks.

### 2.2. Optimization of the Analytical Method for NDMA in Water

The analytical method used in this study was an improved modification of those in previous reports in which dry air was not used to remove water from the cartridge [6,25,26], and secondary amines, including DMA, were not removed prior to chemical derivatization [22]. Optimizations were carried out to dry the sorbent cartridge, select the extracting solvents, and remove secondary amines. A 500 mL water sample spiked with 2 μL of a surrogate (NMBA, 5 mg/L) was passed through a cartridge containing 2.0 g of Carboxen® 572 adsorbent at a flow rate of 10 mL/min. The cartridge was mounted on a vacuum manifold (Visiprep™ SPE vacuum manifold, Sigma-Aldrich) and dried under vacuum (−30 kPa) with and without a silica gel trap (16 g in an impinger) for 60 min. Another drying method used a 1-L/min nitrogen gas flow through the cartridge for 60 min. NDMA in the adsorbent was eluted with 15 mL of a solvent system at a flow rate of 10 mL/min. Two eluent options were examined: (1) a mixture of DCM and methanol (95:5, v/v) and (2) DCM only. Each experimental setting was repeated three times. After the above optimizations were conducted, the duration required for adsorbent drying was tested at 10, 30, 60, and 90 min.

DMA removal efficiency was tested by spiking 0.5 μL of its stock solution (1000 mg/L) or its 1 mL concentrate into the 15 mL eluate. In the former method, the 15 mL eluate was transferred to a 40 mL vial, to which 3 mL of a 1 N HCl solution was added. The mixture was vigorously shaken for 10 min using a mechanical shaker (SR-2DS, Taitec; Koshigaya, Japan). The organic layer was separated and concentrated to 1 mL using a gentle nitrogen gas stream after drying over approximately 0.5 g of $Na_2SO_4$. In the second method, the 15 mL eluate in a centrifuge tube was concentrated to 1 mL using a mild nitrogen gas stream, and 1 mL of 1 N HCl was then added to the concentrate.

The mixture was shaken for 5 min using a Maxi Mix II vortex mixer (Barnstead Thermolyne, Dubuque, IA, USA). The organic phase was separated and dried over approximately 50 mg of $Na_2SO_4$. Both sets of experiments were performed in triplicate.

In a centrifuge tube, 150 μL of the denitrosating reagent was added to each 1 mL concentrate, and the resulting mixture was vortex-mixed for 10 seconds. The mixture was heated at 40 °C for 30 min and concentrated to dryness using a nitrogen gas stream. The pH 10.5 buffer solution (150 μL) and the dansylating reagent (150 μL) were added to the concentrate, and the mixture was shaken for 10 s using the vortex mixer. After the centrifuge tube was heated at 40 °C for 30 min, the mixture was mixed with 50 μL of reagent water. Forty microliter of the analytical sample was injected into the HPLC-FLD system. The instrumental conditions are presented in Table 1.

**Table 1.** High-performance liquid chromatography-fluorescence detection (HPLC-FLD) conditions.

| Parameter | Model/Condition |
| --- | --- |
| Pump | 515 HPLC pump (Waters Co., Milford, MA, USA) |
| Sample Injection | 717 Plus Autosampler (Waters) |
| Stationary Phase | Skypak C18 (4.6 mm × 250 mm × 5 μm, SK Chemicals, Seongnam, Gyeonggi-do, Korea) |
| Mobile Phase | Water:$CH_3CN$ (45:55, v/v) |
| Detector | 474 Fluorescence detector (Waters) |
| Excitation and Emission Wavelengths | 340 nm and 530 nm |
| Injection Volume | 40 μL |
| Flow Rate | 1 mL/min |

*2.3. Method Validation*

The optimized method was validated for the method detection limit (MDL), the limit of quantitation (LOQ), the linearity of a calibration curve ($r^2$), accuracy, and precision. The MDL and LOQ were estimated according to the US EPA's procedure [25]. A five-point linear calibration curve was drawn using a set of standards with 2.0, 20, 40, 60, and 80 ng/L concentrations, and the coefficient of determination ($r^2$) was calculated to determine the linearity of the calibration curve. The accuracy of both the percent recoveries and percent errors between experimental and theoretical values at three levels (2.0, 30, and 60 ng/L) was evaluated. Precision was evaluated using repeatability expressed as relative standard deviations (RSDs) of the three replicates at the same levels as the accuracy.

*2.4. Sampling and Analysis of Tap Water Samples*

Tap water samples were collected in 250 mL amber glass bottles containing approximately 25 mg of sodium thiosulfate from water faucets at 41 sites distributed nationwide throughout Korea, including Jeju Island (Figure 1). The sites were classified into seven regional groups: Seoul (five sites); Gyeonggi (five sites), including Incheon; Gangwon (five sites); Chungcheong (eight sites), including Daejeon and Sejong; Jeolla (seven sites), including Gwangju; Gyeongsang (nine sites), including Busan, Ulsan, and Daegu; and Jeju (two sites). Eleven of the 41 sites (26.8%) were supplied with drinking water from water treatment plants equipped with advanced oxidation processes (AOPs), such as ozone and powdered-activated carbon (PAC) treatment, including all Seoul sites, one site in Gangwon, and five sites in Gyeongsang. Samples were collected from the same sites in summer (16 August–29 September 2016) and winter (6 January–9 February 2017). On each visit, free and total residual chlorine concentrations, pH, and water temperature were measured in situ.

Secondary amines were determined using GC-MS according to the method by Park et al. [9]. Dissolved organic carbon (DOC) concentrations were measured using a TOC analyzer (Sievers 5310C, Boulder, CO, USA) after samples were filtered through a 0.45 μm membrane filter. Absorbance values at 254 nm ($UV_{254}$) were determined using a UV-Vis spectrophotometer (UV-9100, Human Co., Seoul, Korea). Specific UV absorbance values at 254 nm ($SUVA_{254}$) were calculated by dividing $UV_{254}$ values

by DOC concentrations. In addition, total nitrogen concentrations were measured using the persulfate oxidation method [32]. Nitrate concentrations were determined using ion chromatography.

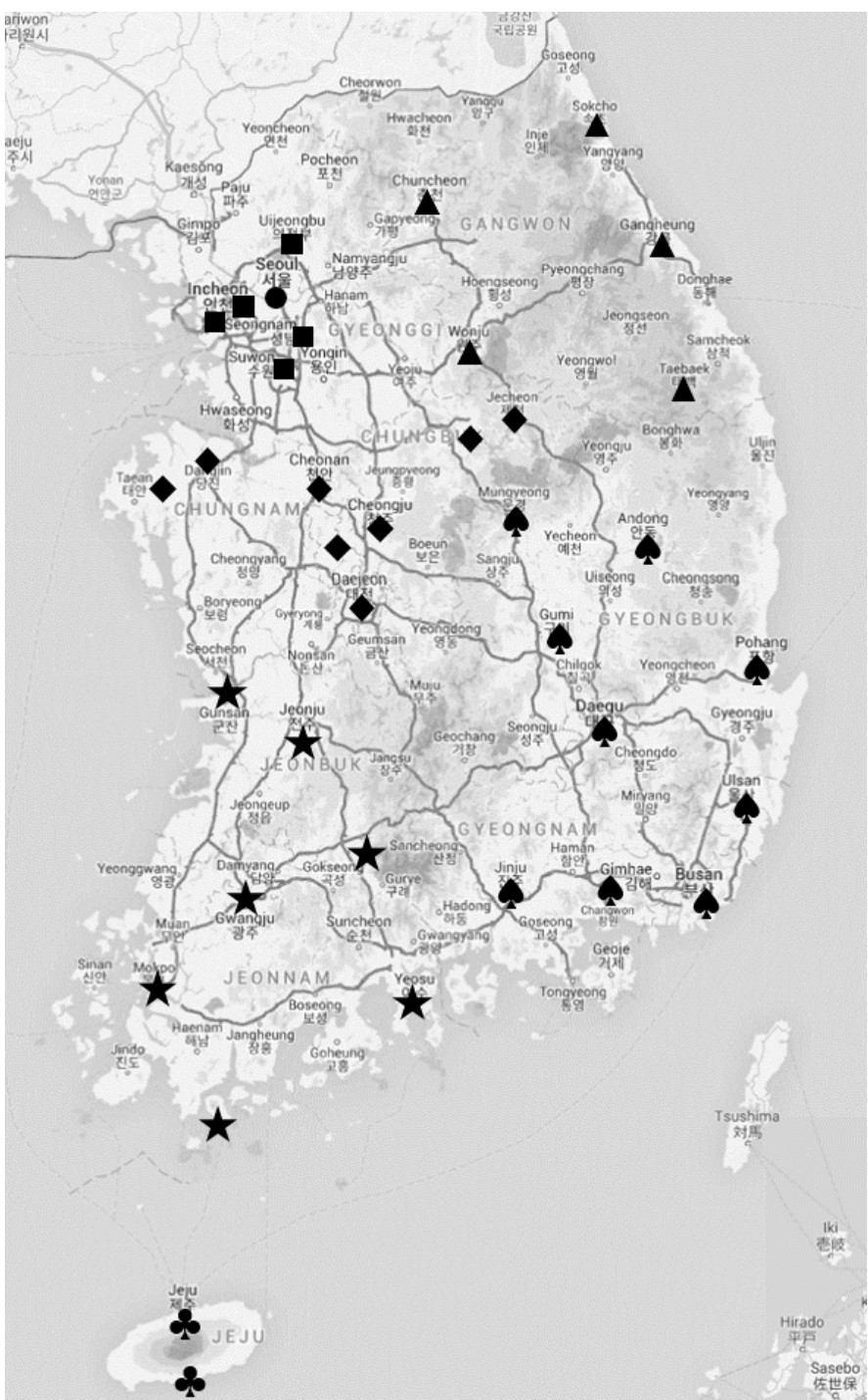

**Figure 1.** Tap water sampling sites. A total of 41 sites [five in Seoul (●), five in Gyeonggi (■), five in Gangwon (▲), eight in Chungcheong (◆), seven in Jeolla (★), nine in Gyeongsang (♠), and two in Jeju (♣)] were visited once in summer and once in winter.

## 2.5. Data Analysis

Data analyses were conducted using IBM SPSS 24 software (Armonk, NY, USA), and a significance level of 5% was used to determine the statistical significance of the tests. A paired *t*-test was

conducted to examine differences in NDMA concentrations between summer and winter. Regional variations of NDMA concentrations among the seven regions were tested using a one-way analysis of variance (ANOVA). Causal relationships between water quality parameters (DMA concentrations; free, combined, and total residual chlorine concentrations; pH; water temperature; DOC concentration; SUVA$_{254}$; total nitrogen concentration; and nitrate concentration) and NDMA concentrations were examined by multiple regression analyses.

## 3. Results and Discussion

### 3.1. Method Optimization

Figure 2 illustrates peak area ratios for six different combinations of three sorbent drying methods and two eluents. Vacuum drying using an in-line silica gel trap and subsequent elution with DCM (first set) indicated the highest chromatographic response; therefore, this set was selected as the first priority for the method. The value for the use of the mixture (5% methanol in DCM, the second set) was approximately half that of DCM only. Vacuum drying without a silica gel trap (third and fourth sets) resulted in lower values than those of vacuum drying with a silica gel trap (first and second sets), probably because of the incomplete drying of the adsorbent because of the withdrawal of atmospheric moisture into the cartridge. This can be explained by the dissolution of NDMA on the adsorbent with an extracting solvent. Adsorbed NDMA needs to be in contact with an eluent to be extracted. If water remains on the surface of the adsorbent, the eluent will not readily contact NDMA. This is because DCM is not readily mixed with water, which may lead to the incomplete extraction of NDMA. To improve the polarity of the eluent, methanol (5%) was added. However, this was not effective; rather, it decreased extraction efficiency possibly because the addition of methanol decreased the solubility of NDMA.

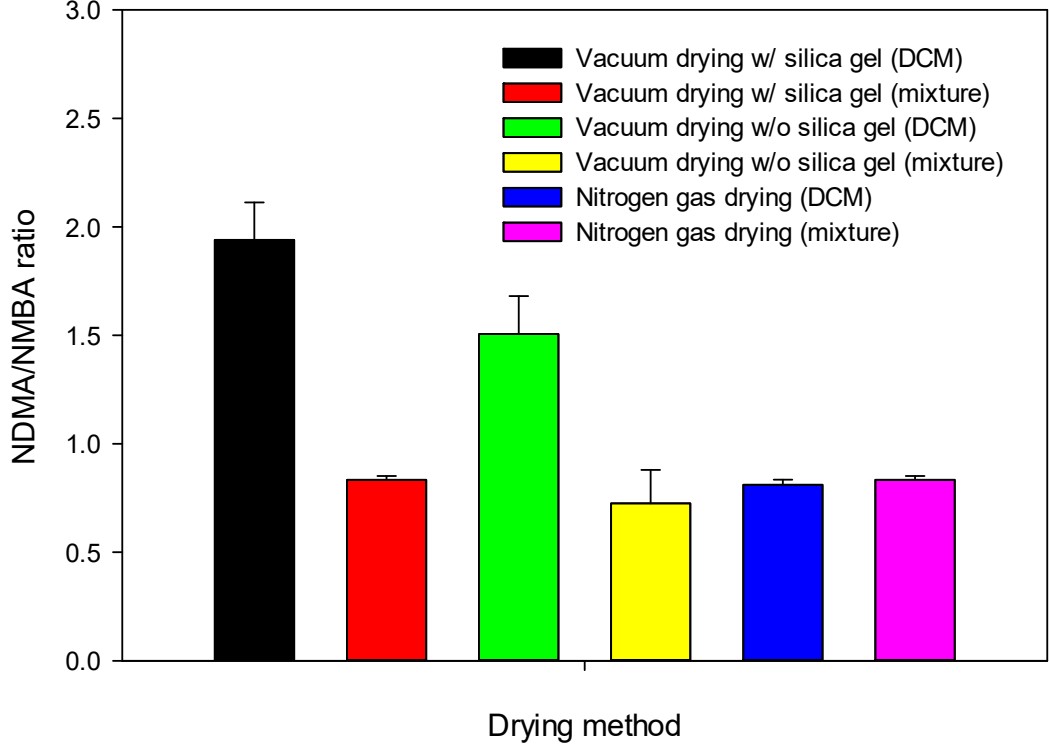

**Figure 2.** Comparison of the *N*-nitrosodimethylamine (NDMA) ratios with NMBA (a surrogate) peak areas among six different combinations of drying methods and eluent options (*n* = 3).

Drying with a nitrogen gas stream (fifth and sixth sets) resulted in area response ratios approximately half that of vacuum drying with a silica gel trap using DCM as an eluent and ratios similar to the second and fourth methods. This suggests a loss of NDMA during drying. Other flow rates and drying durations using a nitrogen stream were attempted, but there was no improvement. Some previous studies used a nitrogen gas flow for adsorbent drying but did not report information regarding the gas flow rate or pressure [14,21].

Therefore, it is not easy to obtain reproducible experimental results without optimizing the two parameters. For instance, McDonald et al. [21] found a relatively low percent recovery of approximately 50% and 79% by using ultrapure and tap water as sample matrices at 10 and 100 ng/L levels, respectively. This suggests that in the eluate, NDMA might be incompletely extracted from the adsorbent because of incomplete drying, or it might be lost prior to sample elution because of excessive drying.

Of the three drying durations, 10 min was found to be optimal (Figure 3). No significant difference between 10 and 30 min was observed, and longer drying periods (60 and 90 min) led to a decrease in peak area ratios, likely because of NDMA loss as a result of the air stream purge. Furthermore, if the drying time is too long, the additional formation of NDMA on the adsorbent can occur because of the reaction of DMA with $N_2O$, nitrosamine ($H_2N_2O$), or $NH_2OH$ [23,24]. Among the activated carbons tested, Carboxen 572, which was adopted in this study, exhibited the least NDMA formation. Therefore, care needs to be taken to produce reliable measurement results during the sample preparation stage.

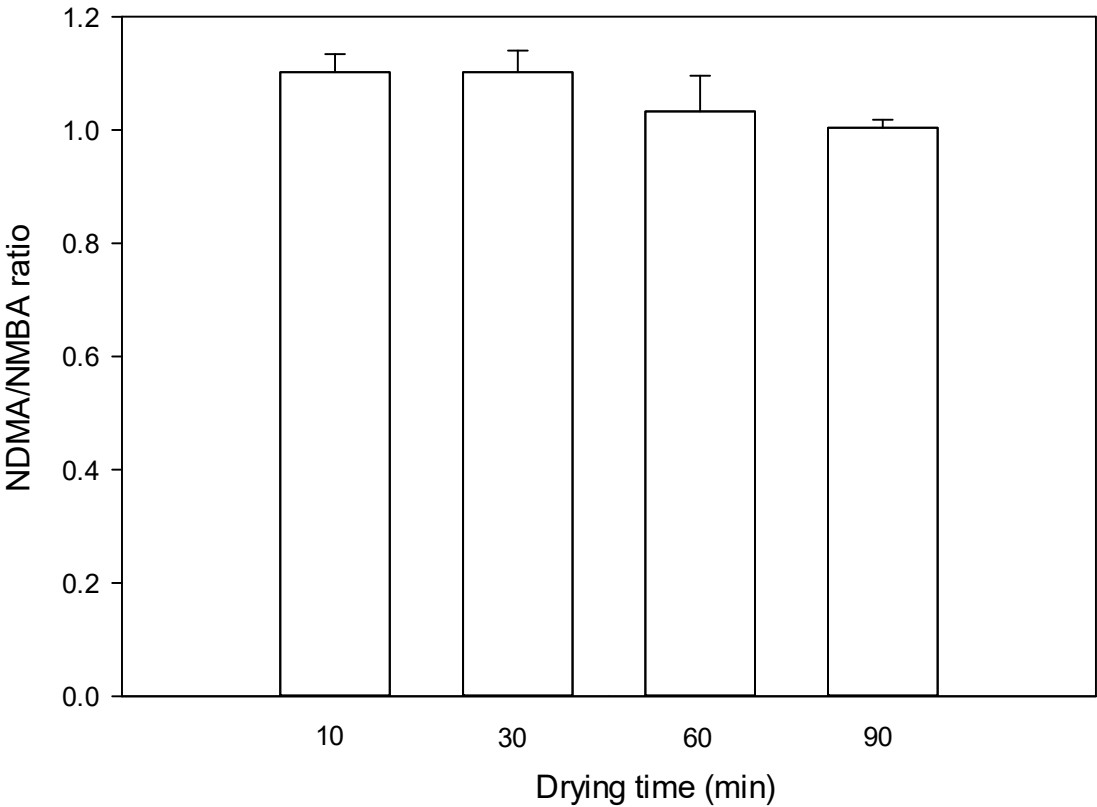

**Figure 3.** Optimization of drying duration for vacuum drying with a silica gel trap (*n* = 3).

The removal of DMA from water by washing the eluate with dilute HCl solution was evaluated, and the result is illustrated in Figure 4. Approximately 70% of DMA remained in the final analytical sample without the acid washing step. To the contrary, approximately 65% of DMA was removed after washing the 15 mL DCM eluate with a 1 N HCl solution, whereas complete removal was attained by washing the 1 mL DCM concentrate with a 1 N HCl solution. This finding suggests that acid washing

is necessary for DMA removal, and the most efficient removal can be achieved by washing a small volume of the concentrate with the acid.

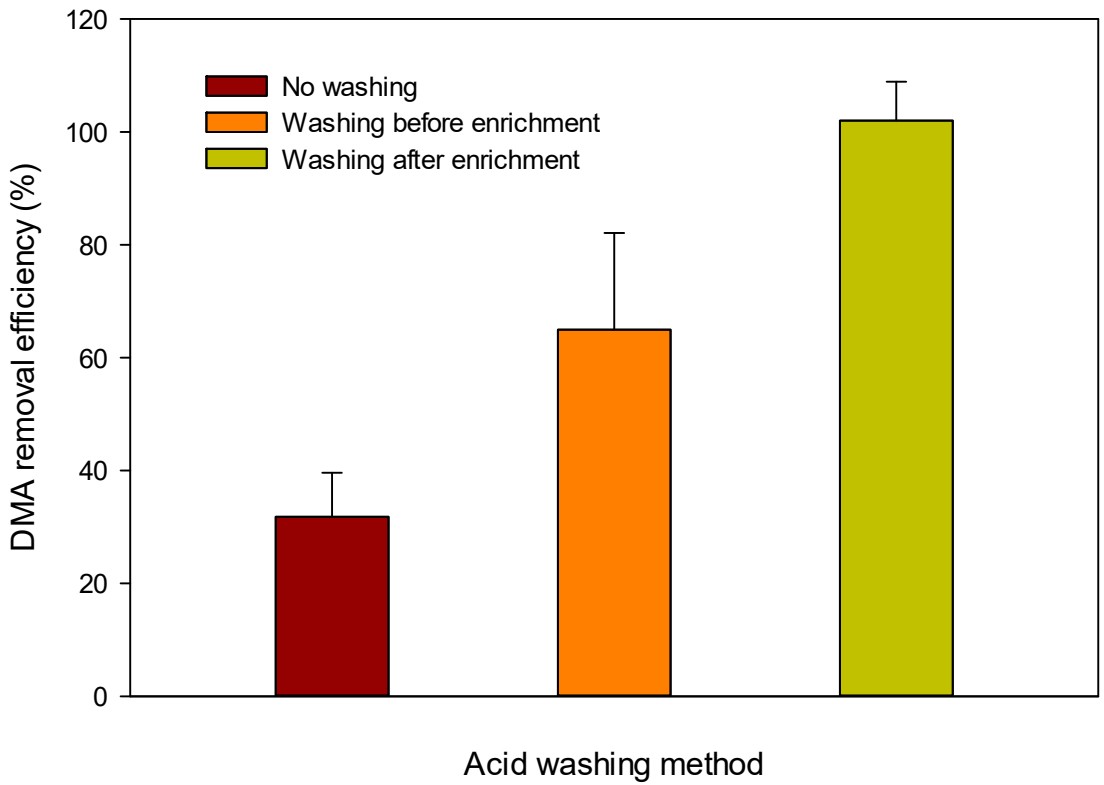

**Figure 4.** Comparison of the removal efficiency of dimethylamine (DMA) between two acid washing options. No acid washing resulted in a DMA removal of only 31.8 ± 7.8%. Washing the 15 mL eluate with a 1 N HCl solution prior to concentrating with a nitrogen gas stream resulted in the removal of 65.0 ± 17.1% of DMA. Concentrating the eluate to 1 mL and then washing it with 1 N HCl resulted in the complete removal (102.2 ± 6.9%) of DMA, indicating no overestimation of NDMA concentration because of the presence of DMA in water.

The final established method based on the above procedures is as follows. A 500 mL water sample was passed through a preconditioned cartridge (20 mL each of hexane, DCM, and methanol) containing 2.0 g of Carboxen® 572 (20–45 mesh) at a rate of 10 mL/min. The upper end of the cartridge was connected to a trap containing ~25 mL (~16 g) silica gel, and vacuum (−30 kPa) was applied for 10 min to dry the adsorbent. DCM was passed through the cartridge at 10 mL/min to elute NDMA and NMBA. The eluate was concentrated to 1 mL by a mild stream of nitrogen gas. One microliter of the 1 N HCl solution was added to the concentrate, and the mixture was vigorously shaken for 1 min using the vortex mixer. The bottom DCM layer was transferred to a 4 mL vial, to which ~50 mg of $Na_2SO_4$ was added. After the filtration of the mixture through a pipette packed with a small amount of defatted cotton, the denitrosating reagent (150 μL) was added, and the mixture was heated at 40 °C for 30 min. The denitrosating reagent and DCM were evaporated using a nitrogen gas stream, and a buffer solution (pH 10.5, 150 μL) and a dansylating reagent (150 μL) were then added to the residue. The mixture was heated at 40 °C for 30 min, and 50 μL of ultrapure water was added. Thereafter, the aliquots of 40 μL were injected for HPLC analysis.

*3.2. Results of Method Validation*

Table 2 presents the results of the method detection limit, limit of quantitation, linearity, accuracy (percent recovery and percent error), and precision (repeatability). The MDL was estimated to be 0.26 ng/L, which is comparable to those from previous studies (0.45 ng/L [21] and 0.28 ng/L [25]),

in which GC-tandem MS was used for instrumental analysis. The calibration curve was linear with a coefficient of determination ($r^2$) of 0.9984. Percent recoveries and errors ranged from 81.1% to 92.1% and from 2.13% to 12.0%, respectively. The percent recoveries are similar to those of the EPA method 521 (83.7 to 94.7% for reagent water and drinking water fortified with NDMA) [25]. Relative standard deviations calculated at three levels were lower than 7.1%, which were also similar to those of the EPA method 521 (4.4 to 12%) [25]. The above results indicate that the current analytical method was validated for the analysis of NDMA in field water samples.

**Table 2.** Method validation results for the determination of NDMA in water.

| Parameter | | Working Concentration (ng/L) Tested | Value |
|---|---|---|---|
| Method Detection Limit | | | 0.26 ng/L |
| Limit of Quantitation | | | 0.82 ng/L |
| Linearity ($r^2$) | | 2.0–80 | 0.9984 |
| Accuracy | Percent Recovery | 2.0 | 89.5% |
| | | 30 | 81.1% |
| | | 60 | 92.1% |
| | Percent Error | 2.0 | 12.0% |
| | | 30 | 11.3% |
| | | 60 | 2.13% |
| Precision | Repeatability (RSD) | 2.0 | 7.09% |
| | | 30 | 3.98% |
| | | 60 | 2.61% |

*3.3. Nationwide Distributions of NDMA Concentrations in Tap Water*

NDMA was found in most samples with detection rates of 95.1% (39 of 41) and 100% in summer and winter, respectively. The overall average (±standard deviation) NDMA concentration was 46.6 (±22.7) ng/L, in the range of 0.13–80.7 ng/L (Table 3).

**Table 3.** Regional and seasonal distributions of NDMA concentrations (ng/L) in nationwide tap water samples.

| Sampling Region | Summer | | Winter | | Both Seasons | |
|---|---|---|---|---|---|---|
| | *n* | Concentration | *n* | Concentration | *n* | Concentration |
| Seoul | 5 | 21.8 ± 12.8 (0.13–32.9) | 5 | 70.2 ± 8.3 (60.2–77.8) | 10 | 46.0 ± 27.5 (0.13–77.8) |
| Gyeonggi | 5 | 46.9 ± 23.5 (18.0–75.4) | 5 | 46.2 ± 25.8 (12.7–70.8) | 10 | 46.6 ± 23.3 (12.7–75.4) |
| Gangwon | 5 | 20.7 ± 7.8 (11.3–29.1) | 5 | 52.5 ± 24.7 (19.3–79.3) | 10 | 36.6 ± 24.1 (11.3–79.3) |
| Chungcheong | 8 | 30.5 ± 23.7 (3.00–66.2) | 8 | 62.7 ± 12.7 (46.7–80.0) | 16 | 46.6 ± 24.8 (3.00–80.0) |
| Jeolla | 7 | 56.3 ± 9.4 (42.8–70.2) | 7 | 63.7 ± 8.4 (54.1–80.7) | 14 | 60.0 ± 9.4 (42.8–80.7) |
| Gyeongsang | 9 | 36.9 ± 16.7 (11.9–58.6) | 9 | 54.9 ± 21.4 (17.2–78.7) | 18 | 45.9 ± 20.8 (11.9–78.7) |
| Jeju | 2 | 2.49 ± 3.30 (0.13–4.86) | 2 | 8.02 ± 5.27 (4.29–11.7) | 4 | 5.26 ± 4.81 (0.13–11.7) |
| Total | 41 | 36.5 ± 20.5 (0.13–75.4) | 41 | 56.2 ± 20.9 (4.29–80.7) | 82 | 46.6 ± 22.7 (0.13–80.7) |

Previous studies demonstrated that NDMA concentrations varied significantly depending on the study region, sampling location (plant effluent/distribution system), season, source water (groundwater/surface water), and disinfectant (chlorine/chloramines), ranging from <MDL to 630 ng/L [1,7,11–14,33–36]. NDMA levels up to 189 ng/L were reportedly found in China [7] but were mostly within 50 ng/L [35,36]. These levels are similar to those observed in this study.

Most of the NDMA concentrations measured in this study exceeded the drinking water concentration (7 ng/L) corresponding to the $10^{-5}$ cancer risk from oral exposure estimated by the EPA IRIS [3]. Thirty-seven (90.2%) and 40 (97.6%) samples exceeded this value in summer and winter, respectively. Moreover, two samples (4.88%) in summer and 13 samples (31.7%) in winter had concentrations greater than 70 ng/L ($10^{-4}$ risk level). Therefore, residents of the Korean Peninsula might undergo excessive cancer risks from the ingestion of chlorinated tap water with/without boiling because NDMA is thermally stable, and its concentration would increase after a long boiling period because of decreased water volume [37].

The NDMA concentrations measured for tap water in this study (<0.26–80.7 ng/L) were much higher than those measured for finished water (<0.50–13 ng/L) at drinking water treatment plants in Korea from 2013 to 2015 [13]. This is probably because NDMA formation continues to occur throughout the distribution pipes after the water is disinfected in the treatment plants [14,38]. Average (±standard deviation) distances from each treatment plant to each sampling site were estimated to be 13.2 (±14.1) km with a range of 0.60 to 66.5 km, leading to various NDMA formations.

*3.4. Regional Variations of NDMA Concentrations*

Table 3 presents NDMA concentrations in the seven regions. The concentrations in the Korean Peninsula for the entire data set ranged from <0.13 (MDL) to 80.7 ng/L, whereas those in the Jeju region were much lower, ranging from <0.13 to 11.7 ng/L. Table 4 shows the regional and seasonal distributions of water quality parameters in nationwide tap water samples. The lower NDMA levels in the Jeju region could be attributed to the use of spring water as a source of drinking water, which might contain lower levels of organic precursors to NDMA (Table 4).

The one-way ANOVA conducted to determine the differences in mean NDMA concentrations among all the data of the seven areas indicated a significant difference ($p = 0.024$) at the 5% significance level. A subsequent post hoc analysis using Tukey HSD indicated a regional difference in the mean NDMA concentration only between the Jeju (the lowest concentration) and the Jeolla (the highest concentration) areas. Significant differences among the other six regions were not found. Excluding the Jeju data, the ANOVA was conducted separately for summer and winter data. A significant difference in the mean NDMA concentrations for the summer data was observed between the Seoul (21.8 ng/L) and Jeolla (56.3 ng/L) regions ($p = 0.019$; Tukey HSD). However, such a difference was not found for the winter data. The above results indicate that average NDMA concentrations in the Korean Peninsula were observed at relatively high levels, but concentrations did not significantly differ among the sampling sites.

**Table 4.** Regional and seasonal distributions of water quality parameters in nationwide tap water samples.

| Sampling Region | Season | DMA (µg/L) | Free Cl (mg/L) | Combined Cl (mg/L) | pH | Water Temp. (°C) | DOC (mg/L) | SUVA (L/mg-m) | Total Nitrogen (mg/L) | Nitrate (mg/L) |
|---|---|---|---|---|---|---|---|---|---|---|
| Seoul | S * | 0.57 ± 0.11 | 0.90 ± 0.44 | 0.04 ± 0.05 | 6.44 ± 0.06 | 29.1 ± 0.9 | 1.16 ± 0.65 | 1.37 ± 0.29 | 8.53 ± 1.82 | 7.27 ± 0.32 |
| | W ** | 0.63 ± 0.19 | 0.39 ± 0.18 | 0.18 ± 0.03 | 6.94 ± 0.04 | 9.42 ± 2.08 | 1.43 ± 0.37 | 0.78 ± 0.35 | 9.61 ± 3.56 | 10.7 ± 0.88 |
| Gyeong-gi | S | 0.81 ± 0.22 | 0.80 ± 0.28 | 0.05 ± 0.04 | 6.81 ± 0.17 | 28.1 ± 2.8 | 1.21 ± 0.19 | 1.59 ± 0.23 | 8.15 ± 0.48 | 7.11 ± 0.94 |
| | W | 0.93 ± 0.40 | 0.69 ± 0.30 | 0.12 ± 0.11 | 7.05 ± 0.04 | 12.0 ± 2.9 | 1.92 ± 0.22 | 0.94 ± 0.31 | 10.8 ± 0.8 | 9.21 ± 0.97 |
| Gangwon | S | 0.92 ± 0.19 | 0.81 ± 0.17 | 0.04 ± 0.03 | 6.86 ± 0.44 | 24.5 ± 2.8 | 1.49 ± 0.61 | 2.10 ± 1.76 | 8.04 ± 2.93 | 6.67 ± 4.21 |
| | W | 0.45 ± 0.10 | 0.69 ± 0.10 | 0.04 ± 0.02 | 7.07 ± 0.27 | 10.6 ± 1.6 | 1.31 ± 0.63 | 1.50 ± 0.47 | 7.28 ± 1.64 | 5.46 ± 2.38 |
| Chung-cheong | S | 0.94 ± 0.37 | 0.68 ± 0.36 | 0.05 ± 0.04 | 6.95 ± 0.46 | 28.9 ± 1.4 | 1.68 ± 1.03 | 1.57 ± 0.29 | 7.30 ± 1.98 | 6.02 ± 2.53 |
| | W | 0.86 ± 0.22 | 0.46 ± 0.32 | 0.14 ± 0.11 | 7.00 ± 0.27 | 11.5 ± 3.1 | 1.39 ± 0.49 | 1.22 ± 0.61 | 8.41 ± 3.65 | 7.14 ± 4.68 |
| Jeolla | S | 0.86 ± 0.18 | 0.90 ± 0.38 | 0.02 ± 0.01 | 6.57 ± 1.82 | 26.0 ± 1.8 | 1.07 ± 0.44 | 1.90 ± 0.57 | 3.73 ± 1.96 | 3.75 ± 2.87 |
| | W | 0.85 ± 0.38 | 0.48 ± 0.16 | 0.07 ± 0.05 | 6.95 ± 2.15 | 10.5 ± 2.6 | 1.20 ± 0.33 | 1.43 ± 0.58 | 5.96 ± 3.45 | 3.90 ± 2.65 |
| Gyeong-sang | S | 0.89 ± 0.66 | 0.92 ± 0.23 | 0.04 ± 0.02 | 6.35 ± 0.35 | 28.1 ± 2.2 | 1.11 ± 0.36 | 1.58 ± 0.52 | 3.25 ± 1.32 | 3.43 ± 2.92 |
| | W | 0.79 ± 0.39 | 0.45 ± 0.31 | 0.10 ± 0.03 | 7.56 ± 0.25 | 8.62 ± 2.40 | 1.83 ± 0.65 | 1.97 ± 2.69 | 5.96 ± 3.45 | 8.21 ± 4.76 |
| Jeju | S | 0.45 ± 0.01 | - | - | - | - | - | - | - | 7.42 ± 0.93 |
| | W | 0.57 ± 0.15 | - | - | - | - | - | - | - | 7.35 ± 0.79 |

*: Summer. **: Winter. -: The data are missing.

### 3.5. Seasonal Variations of NDMA Concentrations

Nationwide water samples were grouped into two seasons (summer and winter) and seven locations (Seoul, Gyeonggi, Gangwon, Chungcheong, Jeolla, Gyeongsang, and Jeju) (Table 3). The average NDMA concentrations in summer and winter were 36.5 and 56.2 ng/L, respectively. The Jeolla and Seoul regions had the highest concentrations of 56.3 and 70.2 ng/L in summer and winter, respectively, whereas Jeju recorded the lowest average concentrations of 2.49 and 8.02 ng/L in summer and winter, respectively. The average NDMA concentrations were higher in winter than in summer, except for the Gyeonggi region, where the concentrations were similar in both seasons (46.9 in summer vs. 46.2 ng/L in winter).

The paired *t*-test conducted for the seasonal concentration comparison of all the samples indicated significantly greater concentrations in winter than in summer with $p = 0.000$. This result indicates that NDMA formation is favored in colder months as opposed to warmer ones. The aforementioned outcome is in agreement with those of previous studies in which the concentrations of NAs, including NDMA, in drinking water samples in colder seasons (winter and fall) were higher than those in summer [11,18,38,39].

Higher NDMA concentrations in colder seasons can be explained by lower microbiological activity and lower photodegradation because of less sunlight [38,40]. Since the microbial activities of *Nitrosomonas* and *Nitrobacter* are very low in colder seasons, ammoniacal nitrogen is not readily oxidized to more highly oxidized forms, such as nitrite and nitrate [40,41]. Therefore, higher levels of ammoniacal nitrogen in winter than those in summer might cause chloramines to form at a higher rate [42], followed by the ready formation of NDMA via the pathway involving unsymmetrical dimethylhydrazine chloride (UDMH-Cl) [43].

### 3.6. Relationship between NDMA Concentrations and Water Quality Parameters

A multiple regression analysis was conducted for the entire data set (with the exception of Jeju) by setting the NDMA concentration as the dependent variable and all other water quality parameters (including DMA concentration, free residual chlorine, combined residual chlorine, total residual chlorine, pH, water temperature, DOC, $SUVA_{254}$, nitrate concentration, and total nitrogen concentration) as independent variables. Only a single parameter, water temperature, was included in the regression equation with $p = 0.000$ and $R^2 = 0.271$ (adjusted $R^2 = 0.261$) (Figure 5). A relatively small difference between $R^2$ and adjusted $R^2$ indicates that the following equation is a good description of NDMA formation:

$$NDMA = 71.7 - 1.30 \times \text{water temp}$$

Additionally, there was no problem regarding multicollinearity since variance inflation factor (VIF) values were below 10 (1.000–1.352) for all dependent variables.

The above equation indicates the favorable formation of NDMA at low water temperatures. However, other parameters were not significantly related to NDMA concentration ($p > 0.05$). This is in good agreement with previous studies, in which high NDMA formation was observed in colder seasons [11,18,38,39].

Multiple regression analyses were also conducted separately for six different regions except Jeju. The following significant regression equation was obtained only for the Jeolla data:

$$NDMA = 13.7 + 55.4 \times DMA + 5.84 \times SUVA + 80.7 \times \text{Combined Cl}$$

The coefficient of determination ($R^2$) for this regression equation was high at 0.936 (adjusted $R^2 = 0.917$), and the *p*-values for DMA, SUVA, and combined Cl were 0.000, 0.001, and 0.001, respectively. This is consistent with previous reports: NDMA formation increases as the DMA concentration, $NH_2Cl$ concentration [44], and SUVA [45] increase.

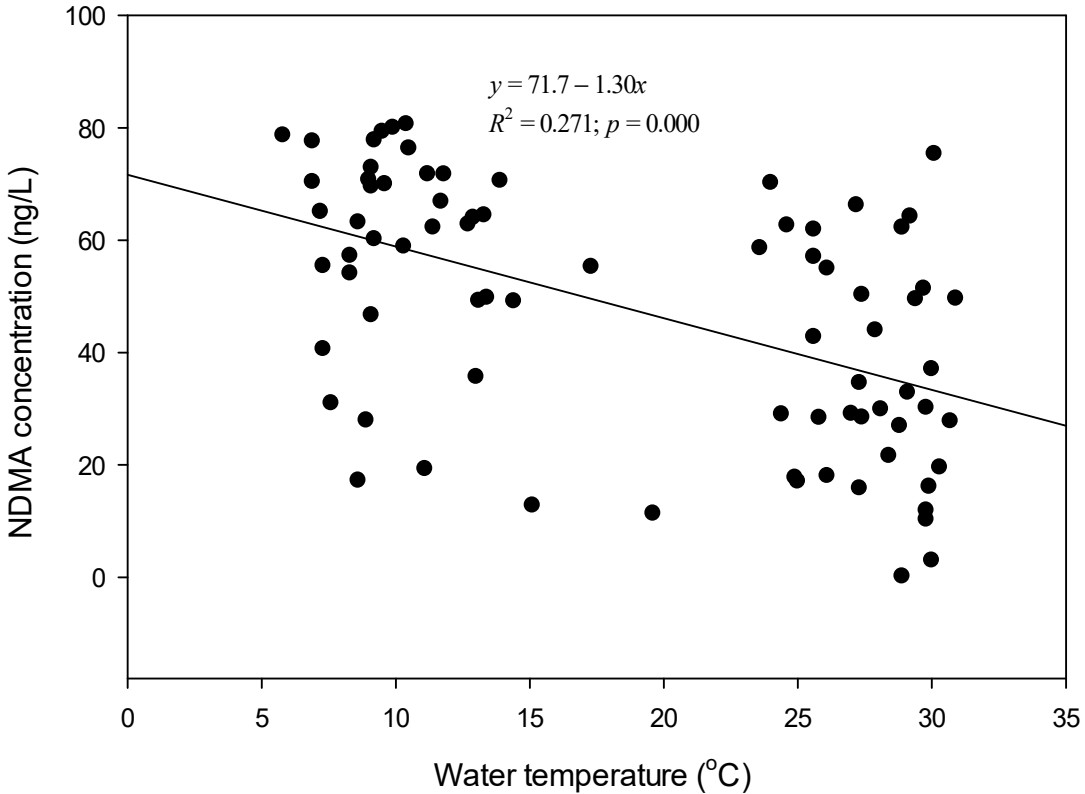

**Figure 5.** The relationship between water temperature and NDMA concentration. The two variables were significantly but weakly related to each other with $R^2 = 0.271$ and $p = 0.000$.

Some water treatment plants employ AOPs, including ozone treatment and PAC, as mentioned in Section 2.4. The variations in treatment processes could be one of the reasons for the weak relationship between NDMA concentration and other field water parameters. Therefore, excluding AOP data, multiple regression indicated a slightly increased $R^2$ (0.312) and adjusted $R^2$ (0.285) with two independent variables, such as water temperature and total chlorine concentration ($n = 56$). The following regression model is suggested for the data:

$$NDMA = 61.8 - 1.43 \times \text{water temp} + 17.4 \times \text{T-Cl}$$

Only a few of the parameters considered above were found to be significantly related to NDMA formation, suggesting that those factors complicatedly act on the reaction. Moreover, other parameters in addition to those listed above could be involved in the formation of NDMA. In the future, elegantly designed laboratory studies are necessary to determine such parameters.

The NDMA concentration is expected to decrease in the future in the authors' country because NDMA in treatment plant water should be quarterly monitored since 2018, and livestock raising is gradually modernized to minimize the discharge of animal wastes, one of its major organic precursor.

## 4. Conclusions

An analytical method using HPLC-FLD for the quantitative determination of NDMA in drinking water was optimized for SPE cartridge drying methods and acid washing for the removal of DMA from water. Vacuum drying through silica gel and acid-washing of the enriched eluate produced satisfactory method validation results.

NDMA concentrations were significantly higher in winter than in summer, suggesting that in colder seasons, NDMA formation is favored, whereas NDMA degradation occurs less favorably. This observation was confirmed by the multiple regression analysis of the entire data set, where water

temperature was an important parameter included in the regression equation. NDMA concentrations varied significantly depending on sampling sites (except for the Jeju region), and they were distributed at high levels with an average of 47 ng/L and a maximum of 81 ng/L. Therefore, the oral consumption of NDMA in drinking water may pose excessive cancer risks for the residents of the Korean Peninsula.

This study is not without its limitations. The sample size was not large enough to observe statistical significance for the tests conducted, although samples were obtained from sites nationwide. Moreover, samples were only obtained from faucets. If a study on the NDMA concentrations throughout the distribution systems is conducted, more useful information regarding its formation could be obtained. Furthermore, studies to elucidate the sources of the high levels of NDMA and how its formation can be controlled are necessary to prevent possible health risks in these regions. Because relatively hydrophilic NDMA is not readily removed by the AOP, one of the favored method would be to reduce its formation by removing its organic precursor, DMA. This research is ongoing in the authors' laboratory, and the result will be published shortly.

**Author Contributions:** Conceptualization, H.K.; methodology, H.K., S.P. and S.J.; software, H.K. and S.P.; validation, S.P.; formal analysis, S.P. and S.J.; investigation, H.K.; resources, H.K., S.P. and S.J.; data curation, S.P. and H.K.; writing—original draft preparation, S.P.; writing—review and editing, H.K.; visualization, S.P.; supervision, H.K.; project administration, H.K.; funding acquisition, H.K.

**Funding:** This work was supported by the National Research Foundation of Korea (NRF) grant from the Korea government (MSIT) (No. 2015R1A2A203008216).

**Acknowledgments:** The authors would like to thank Enago (www.enago.co.kr) for the English language review. The authors are also grateful to assistant editor and reviewers for their insightful comments and suggestions.

**Conflicts of Interest:** The authors declare no conflict of interest.

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
