# Peer review of "Regional and Seasonal Distributions of N-Nitrosodimethylamine (NDMA) Concentrations in Chlorinated Drinking Water Distribution Systems in Korea"

_water, doi:10.3390/w11122645_

Round 1

Reviewer 1 Report

Thank you for the opportunity to review this interesting manuscript.

I believe that the scientific quality of the study has increased, and I have only a minor suggestion for the authors.

In the introduction section, the panorama of the NDMA diffusion has been implemented, but I think it could be of interest to insert a brief mention of the problem of the disinfectants by-products in the drinking water distribution.

For example, in Europe (especially in Italy) we had serious problems with the presence of by-products in the drinking water (see, for example: 1) Azara A, et al. Derogation from drinking water quality standards in Italy according to the European Directive 98/83/EC and the Legislative Decree 31/2001 - a look at the recent past. Ann Ig 2018;30:517-26. doi:10.7416/ai.2018.2252; 2) Dettori, M.et al. Population Distrust of Drinking Water Safety. Community Outrage Analysis, Prediction and Management. Int. J. Environ. Res. Public Health 2019, 16, 1004).

I suggest the authors describe this public health issue, focusing the attention on the interest to investigate and monitoring the presence of the by-products, in order to strengthen the research question of the study. 

As I believe that the study could be of interest to the readers, I hope the authors will follow the suggestions.

Reviewer 2 Report

Dear Authors,

Here are the comments and suggestions that are necessary to improve the manuscript

In my opinion, it would be valuable to add information on whether the concentrations of NDMA is known (from literature) - whether it is expected that the concentration will rise / fall in the coming years. Valuable information is also whether samples were collected in one and the same year, or were collected over several years. I would suggest to add this information to the manuscript (in the methods and in the introduction/ discussion).

I would also recommend to correct the adjustment settings in table 2 - it is difficult to read it in its current form.

In Table 3 –in the head of the table -“Total concentration” is mentioned– I think it is Mean Concentration

Table 4 has no citation in the main text ; there is also a lack of description of the results presented there (and of its discussion).  Please add the necessary information.

There is: “The lower NDMA levels in the Jeju region could be attributed to the use of spring water as a source of drinking water, which might contain lower levels of organic precursors to NDMA (Table 5)”. – there is no explanation of table 5 in the manuscript – please add the necessary information (the short description of data placed in table 4 and 5 is needed).

 I suggest to change the description of the Y axis in Fig. 4,  - instead of “Percent removal (%)” it should be something like: “DMA removal efficiency (%)”.

There is the information that, the NDMA concentrations measured for tap water in the study were much higher than those measured for finished water (lines 296-301). Adding information about the ways / possibilities of reducing NDMA concentration in water to the text would be very informative. I realize that the Authors are aware of that as they have concluded that: “Furthermore, studies to elucidate the sources of the high levels of NDMA and how its formation can be controlled are necessary to prevent possible health risks in these regions”.

To sum up, in my opinion, the manuscript is very interesting. It requires some improvements and deserves to be recommended for editorial process submission.

Author Response

This manuscript is a resubmission of an earlier submission. The following is a list of the peer review reports and author responses from that submission.

Round 1

Reviewer 1 Report

Thank you for the opportunity to review the present manuscript. The study aims at determining NDMA concentration in tap water samples collected in Korea, using HPLC-FLD method. I found the study very interesting and well presented, and the study design appropriated. For these reasons, I think that the study can be of interests to the readers. Thus, I suggest the following minor revisions before publishing the manuscript.   1) Introduction section. Even though the Introduction is clearly written, I suggest the Authors implement the international panorama, focusing the attention on the other observations worldwide, which can better define the study background and the importance of conducting observations on the NDMA presence in all (not only Korean) drinking water supplies.   2) Materials and Methods. This section is well presented, but there is no reference to the standard protocol the Authors have followed for the determination of NDMA concentration. I suggest to clearly refer to the protocol used or to declare the followed method.   3) Results section. Figure 2 and figure 3 captions contain explanations of the methods used and the results obtained. I suggest deleting those explanations, leaving the only description of the figure. Moreover, the colours used in figure 3 confuse the reader: are they necessary?   4) Conclusion section. This section can be written without using a bulleted list. Finally, the Authors should declare the limitations or weaknesses of their study.   I hope the Authors will receive the suggestions and make the corrections because I think that the study will be well received by the scientific community.

Reviewer 2 Report

This study established an analytical method for the determination of NDMA in water using HPLC-FLD including the optimization of sorbent drying conditions and the removal of secondary amines, and then investigated the NDMA concentration in tap water samples in Korea for its regional and seasonal distributions. I recognized the scientific significance of this study and the manuscript was well written using understandable English. However I do not recommend it is considered for publication as it is due to the following reasons.

1) In the establishment of the analytical method, I guess the authors modified the existing method. I do not understand what the existing method is and how they modified it. This is mainly because they did not give any citations in the section 2.2.

2) It is necessary to explain about theoretically the result of the optimization of the analytical method, as illustrated in Figure 2.

3) The results related to the description in Line 190-191 should be provided as a supplement.

4) The description in Line 204-206, which includes both incomplete drying and excessive drying, seems meaningless. I wonder what the authors wanted to mention here.

5) What do they mean “field water samples” in Line 244. I really suspect that the optimized method is directly applicable to samples other than tap water due to possible existence of materials which can affect the analysis.

6) In the method optimization, MDL and accuracy of the established method has to be compared to those of the existing methods. As cited as “previous studies”, I believe there are the methods to be compared.

7) In the section 3.3, the data obtained in this study were compared to those reported previously. I believe the previous reports did not use the optimized method in this study. Is it possible to make a fair comparison?

8) In the section 3.6, I wonder why the authors got the significant regression equation only for the Jeolla data. DMA, SUVA and Cl should affect the NDMA analysis everywhere.

9) If AOP affects the relationship between NDMA concentration and other field water parameters, why didn’t the authors analyze the relationship excluding the data from water treatment plants with AOP?
